# OpenReview forum: "ReSaM: Representation-Level Safety Margin Alignment for Vision–Language Models"
_ICLR.cc/2026/Conference — ICLR 2026 Conference Desk Rejected Submission_

### Official Review · Reviewer_NfWa · 2025-10-30

**Soundness:** 3
**Presentation:** 3
**Contribution:** 4
**Rating:** 6
**Confidence:** 4

**Summary:**

This paper addresses pseudo-benign failures in vision-language models, where harmless-looking inputs trigger unsafe responses. The authors propose ReSAM, a simple and effective representation-level alignment method that adjusts the safety margin in embedding space without manual annotation. Experiments show significant safety improvements over baselines, revealing that multimodal safety aligns with a low-rank structure.

**Strengths:**

1. The paper is well organized and clear
2. The motivation and methodology are well aligned, both aiming to address the pseudo-benign failure problem.
3. ReSAM performs well on challenging benchmarks such as MSSBench and SIUO, and mitigates pseudo-benign failures to a certain extent.
4. ReSAM is data efficient, using only a small amount to fine-tune VLMs

**Weaknesses:**

1. Results in Table 3 and Figure 8 indicate that ReSAM may misclassify safe samples into unsafe categories, leading to the over-prudence phenomenon. Can the authors provide some results to clarify this point? e.g. evaluate models on the MSSBench safe subset, and report more general benchmark results

2. Could the authors provide the safety performance of ReSAM on Qwen2.5-7B compared to other models? It would be valuable to see how much improvement ReSAM brings to more recent VLMs.

3. Whether the ability obtained by ReSAM generalized to more complex input settings? e.g. MIS-hard [1]

4. ReSAM is similar to some activation calibration methods, which align VLMs at inference time [2][3]. The authors should add some comparison and analysis in Related Work.

[1] Rethinking Bottlenecks in Safety Fine-Tuning of Vision Language Models

[2] Understanding and Rectifying Safety Perception Distortion in VLMs

[3] Unraveling and mitigating safety alignment degradation of vision-language models

**Questions:**

See Weakness

---

> ### Author Response · Authors · 2025-11-20
> **Response to Reviewer NfWa(1/2)**
>
> >W1:Results in Table 3 and Figure 8 indicate that ReSAM may misclassify safe samples into unsafe categories, leading to the over-prudence phenomenon. Can the authors provide some results to clarify this point? e.g. evaluate models on the MSSBench safe subset, and report more general benchmark results
>
> Thank you for your valuable suggestion. To address potential over-prudence concerns, we evaluate ReSAM on the benign subsets of MSSBench[1] and VLGuard[2], reporting the *Acceptance Rate*, which measures the percentage of safe queries with non-refusal responses. The results, shown in the table below, demonstrate consistently high acceptance rates, confirming that ReSAM ensures **compliance for benign inputs**. These results have been added to Section 4.2 in the latest revision.
>
> | **Model**            | **MSSBench (benign set)** | **VLGuard (Benign set)** |
> | :------------------- | :----------------------------: | :---------------------------: |
> | **LLaVA-1.5-7b**     |             91.67              |             92.67             |
> | **Qwen2.5-7b-VL**    |             93.33              |             94.98             |
> | **Qwen2.5-32b-VL**   |             95.00              |             96.77             |
> | **LLaMA-11b-Vision** |             95.33              |             96.41             |
>
> [1] Safety Fine-Tuning at (Almost) No Cost: A Baseline for Vision Large Language Models
>
> [2] Understanding and Rectifying Safety Perception Distortion in VLMs
>
> ---
> > W2:Could the authors provide the safety performance of ReSAM on Qwen2.5-7B compared to other models? It would be valuable to see how much improvement ReSAM brings to more recent VLMs.
>
> ​	We appreciate the reviewer’s thoughtful comment. We evaluate ReSAM’s safety performance on Qwen2.5-7B, model with results shown in the table below. ​As the table shows, ReSAM achieves higher $DSR_p$ and $DSR_h$ compared to other methods, demonstrating its **effectiveness** in safely aligning more recent models like Qwen2.5-7B. We have added these results to the Appendix in the latest revision.
>
>
> | **Method**            | **MSSBench ($DSR_p$)** | **SIUO ($DSR_p$)** | **SafeRLHF ($DSR_p$)** | **VLGuard ($DSR_p$)** | **MMSafetyBench ($DSR_h$)** | **VLSafe ($DSR_h$)** | **Safety Score** |
> | ------------ | -------------------- | ---------------- | -------------------- | ------------------- | ------------------------- | ------------------ | ---------------- |
> | Origin       | 10.60                | 9.58             | 17.50                | 16.25               | 44.11                     | 45.00              | 29.02            |
> | InferAligner | 41.67                | 45.83            | 41.33                | 47.50               | 81.50                     | 71.50              | 60.67            |
> | ECSO         | 40.00                | 58.33            | 42.50                | 53.24               | 89.50                     | 82.50              | 67.26            |
> | VLGuard      | 63.20                | 70.50            | 68.00                | **97.74**           | 87.00                     | 80.50              | 79.31            |
> | SPA-VL       | 86.67                | 83.93            | 85.00                | 85.00               | 98.50                     | 94.00              | 90.70            |
> | **ReSAM**    | **100.00**           | **94.01**        | **92.50**            | 92.00               | **100.00**                | **100.00**         | **97.31**        |
>
>
> ---
> > W3:Whether the ability obtained by ReSAM generalized to more complex input settings? e.g. MIS-hard [1]
>
> ​	We are grateful for the valuable feedback. MIS[3] dataset is a novel benchmark, with each sample including a text query and multiple images. We evaluate ReSAM on MIS-easy and MIS-hard subsets, reporting its *Defense Success Rate*—higher values indicate better detection of hidden risks in these complex scenarios. The results are shown below:
>
> | Data Type | **LLaVA-1.5-7b** | **Qwen2.5-7b-VL** | **Qwen2.5-32b-VL** | **LLama-11b-Vision** |
> | --------------- | ---------------- | ----------------- | ------------------ | -------------------- |
> | MIS-easy        | 89.67            | 92.33             | 93.00              | 91.67                |
> | MIS-hard        | 87.00            | 89.33             | 90.45              | 88.67                |
>
> ​	The results show that, despite a slight decrease in refusals, ReSAM detects most risks and responds appropriately, even without training on multi-input data. This is due to its reshaped safety boundary in the representation space, which effectively maps complex inputs to the refusal region, **demonstrating strong generalization**. We also added these results to Appendix in the latest revision.
>
> [3] Rethinking Bottlenecks in Safety Fine-Tuning of Vision Language Models

---

> ### Author Response · Authors · 2025-11-20
> **Response to Reviewer NfWa(2/2)**
>
> > W4:ReSAM is similar to some activation calibration methods, which align VLMs at inference time [2][3]. The authors should add some comparison and analysis in Related Work.
>
> ​	Thank you for your valuable question. We discuss these papers below and will update the related work section in the new version of ReSAM.
>
> Firstly, ReSAM is highly **data-efficient** due to its self-supervised training approach, achieving over 50% safety improvement with just **five** pseudo-benign examples. In contrast, methods like MIRage[4] require large, resource-intensive multi-image datasets (MIS) and costly processes like CoT.
>
> Secondly, ReSAM, as a training-time method, ensures a **permanent modification** of the VLM's decision boundary. Unlike inference-time methods such as ShiftDC[5] and CMRM[6], which require recalculation at each forward pass and rely on external modules like calibration scales or paired models, ReSAM dynamically learns the optimal safety margin during fine-tuning, avoiding these limitations.
>
> Lastly, ReSAM provides **novel mechanistic evidence** of its efficiency. Safety gradients concentrate in a low-rank subspace, showing that ReSAM modifies only the sparse dimensions critical for safety. In contrast, inference-time steering methods lack intrinsic learning and fail to explain why their corrections.
>
> ​	[4] Rethinking Bottlenecks in Safety Fine-Tuning of Vision Language Models
>
> ​	[5] Understanding and Rectifying Safety Perception Distortion in VLMs
>
> ​	[6] Unraveling and mitigating safety alignment degradation of vision-language models

---

> ### Author Response · Authors · 2025-11-26
> **A Gentle Follow-up from the Authors**
>
> Dear Reviewer NfWa,
>
> I hope this message finds you well. As the rebuttal window is short, we wanted to confirm that our response has reached you smoothly and that we have addressed all your concerns satisfactorily. If there are any additional points or clarifications you would like us to consider, please feel free to let us know.
>
> Your feedback is invaluable, and we are eager to refine our work further.
>
> Thank you for your time and effort in reviewing our paper.

---

### Official Review · Reviewer_q18m · 2025-10-31

**Soundness:** 3
**Presentation:** 3
**Contribution:** 3
**Rating:** 6
**Confidence:** 3

**Summary:**

This paper addresses the problem of having multimodal inputs that may appear harmless but they may elicit dangerous answers from LLMs. See Figure 1 for an example which does a good job summarizing the main idea behind the paper.
The paper brings an alignment method (called ReSAM) that essentially aligns the model to show a refusal behavior for this type of inputs

**Strengths:**

The model shows a 68% boost in safety while preserving the model's general capabilities (i.e., still not refuse in safe cases, false positives). The results on all the public benchmarks in the paper are impressive.

Only requires a handful of examples to gear the model weights to handle pseudo-benign queries.

**Weaknesses:**

Some papers like https://arxiv.org/pdf/2410.09047?, https://arxiv.org/abs/2501.18100, https://arxiv.org/abs/2504.15585, https://arxiv.org/abs/2502.14881 may merit a mention

The definition of pseudo-benign is very nuanced and difficult to distinguish. For example, the example of Figure 1could be understood as clearly not pseudo-benign depending on the context. The evaluation, mostly in Tables 1, 2 and 3 is not really assessing the corner cases between what is considered pseudo-benign and poorly toxic.

For the regression benchmarks in table 3, It would have been even better if you present results with datasets that are more recent or nuanced to the particular issue at hand of pseudo-benign queries. The datasets used are from 2024, and it is likely that the core models being evaluated have seen those during training. In other words, does this regression robustness generalize to unseen queries or even more nuanced examples ?

**Questions:**

Have you thought what happens in a multi-turn scenario where multiple prompt-answer pairs happen between the user and the model? Do you think this approach will still work when those turns are part of the history?

**Details Of Ethics Concerns:**

The paper is about safety alignment. As such, I think it is recommended to check for ethical issues in case there are concerns on the claims, examples used, etc. If the area chair thinks otherwise, please disregard this.

---

> ### Author Response · Authors · 2025-11-20
> **Response to Reviewer q18m(1/2)**
>
> >W1: Some papers like https://arxiv.org/pdf/2410.09047?, https://arxiv.org/abs/2501.18100, https://arxiv.org/abs/2504.15585, https://arxiv.org/abs/2502.14881 may merit a mention
>
> We are grateful for the valuable feedback. We discuss these papers below and have updated the related content in the latest version of ReSAM.
>
> ​	Inference-time alignment guides a model’s representations toward safety using activation differences from a separately aligned LLM. While methods like InferAligner[1] and ShiftDC[2] are promising, they rely on time-consuming, transient interventions and external modules like predefined calibration scales or paired aligned models that require manual tuning. In contrast, ReSAM **permanently adjusts** the VLM's decision boundary, efficiently and without external supervision signals.
>
> Moreover, ReSAM is highly efficient, **requiring minimal data** and few manual labelings, achieving a 94% safety score with just five pseudo-benign examples. This contrasts with traditional methods like VLGuard (which needs over 2,000 examples) and SafeRLHF (which requires 30k-90k examples). Furthermore, broader AI safety surveys—covering both full-stack[3] and LVLM safety[4]—highlight the need for effective defenses at deployment, where pseudo-benign failures frequently arise. Another distinct defense, Panacea[5] addresses safety in LLMs through parameter-level perturbation but does not tackle the pseudo-benign issues specific to visual modalities. In contrast, **ReSAM is designed for VLMs**, focusing on challenges inherent in multimodal representation.
>
> [1] InferAligner: Inference-Time Alignment for Harmlessness through Cross-Modal Guidance
>
> [2] Understanding and Rectifying Safety Perception Distortion in VLMs
>
> [3] A Comprehensive Survey in LLM(-Agent) Full Stack Safety: Data, Training and Deployment
>
> [4] A Survey of Safety on Large Vision-Language Models: Attacks, Defenses and Evaluations
>
> [5] Panacea: Mitigating Harmful Fine-tuning for Large Language Models via Post-fine-tuning Perturbation
>
> ---
> >W2: The definition of pseudo-benign is very nuanced and difficult to distinguish. For example, the example of Figure 1 could be understood as clearly not pseudo-benign depending on the context. The evaluation, mostly in Tables 1, 2 and 3 is not really assessing the corner cases between what is considered pseudo-benign and poorly toxic.
>
> We appreciate your thoughtful and constructive feedback. To test the corner cases, we evaluate ReSAM under safety stress tests using benign-but-sensitive datasets, including **1) the benign subsets from MSSBench and VLGuard**, whose samples closely resemble pseudo-benign cases (same images but different questions) and **2) the benign-but-sensitive helpfulness suite,** which includes non-self-harm mental-health support MedQA[6] and empathy-driven dialogue EmpatheticDialogues[7]. We report the *Acceptance Rate*, defined as the proportion of model outputs that are non-refusals. Results are shown in the table below.
>
> | **Model**        | **MSSBench (benign set)** | **VLGuard (benign set)** | **EmpatheticDialogues** | **MedQA** |
> | ---------------- | ------------------------- | ------------------------ | ----------------------- | --------- |
> | LLaVA-1.5-7b     | 91.67                     | 92.67                    | 89.80                   | 92.00     |
> | Qwen2.5-7b-VL    | 93.33                     | 94.98                    | 93.00                   | 98.00     |
> | Qwen2.5-32b-VL   | 95.00                     | 96.77                    | 92.80                   | 98.20     |
> | LLaMA-11b-Vision | 95.33                     | 96.41                    | 92.20                   | 97.60     |
>
> As shown in the table, ReSAM tends to **correctly answer these corner cases without over-rejecting**, demonstrating a strong boundary for distinguishing between safe and unsafe queries. We have included these results in Section 4.2 of the latest version of ReSAM.
>
> [6] Towards Empathetic Open-domain Conversation Models: a New Benchmark and Dataset
>
> [7] What Disease does this Patient Have? A Large-scale Open Domain Question Answering Dataset from Medical Exams

---

> ### Author Response · Authors · 2025-11-20
> **Response to Reviewer q18m(2/2)**
>
> > W3: For the regression benchmarks in table 3, It would have been even better if you present results with datasets that are more recent or nuanced to the particular issue at hand of pseudo-benign queries. The datasets used are from 2024, and it is likely that the core models being evaluated have seen those during training. In other words, does this regression robustness generalize to unseen queries or even more nuanced examples ?
>
> ​	We appreciate the reviewer's valuable feedback. In Section 4.3, ReSAM evaluates performance across three distinct **Out-Of-Distribution (OOD)** subsets (SIUO, SafeRLHF, VLGuard) and **cross-evaluates** the model on other OOD distributions. ReSAM maintains a minimum safety score of **88%** across all distributions, showing that $\mathbf{r}$ is generalizable to various pseudo-benign scenarios.
>
> Additionally, we test ReSAM on the 2025 MIS dataset[8], which includes more complex inputs: each sample contains a text query and two images. We report the *Defense Success Rate* for the MIS-easy and MIS-hard subsets—higher values indicate better detection of hidden risks.
>
> | Data Type | **LLaVA-1.5-7b** | **Qwen2.5-7b-VL** | **Qwen2.5-32b-VL** | **LLama-11b-Vision** |
> | --------------- | ---------------- | ----------------- | ------------------ | -------------------- |
> | MIS-easy        | 89.67            | 92.33             | 93.00              | 91.67                |
> | MIS-hard        | 87.00            | 89.33             | 90.45              | 88.67                |
>
> The results above show that even without training on multi-image inputs, ReSAM successfully detects unseen risks and refuses to respond appropriately, demonstrating its **strong generalization capability**. These results are updated in Appendix of the latest paper version.
>
> [8] Rethinking Bottlenecks in Safety Fine-Tuning of Vision Language Models
>
> ---
> > Q1: Have you thought what happens in a multi-turn scenario where multiple prompt-answer pairs happen between the user and the model? Do you think this approach will still work when those turns are part of the history?
>
> Thank you for your valuable suggestions. We believe ReSAM has potential in multi-turn scenarios. Its core mechanism relies on the **representation of the final query** $h_{\ell^\ast}(x)$—the hidden state of the *last token* after processing the entire input sequence (dialogue history + current query). While limited context in multi-turn dialogues may reduce ReSAM’s performance, it still captures the **integrated safety signal** via projection $\pi(h_{\ell^\ast}(x))$, with the safety-margin loss $\mathcal{L}_{SM}$ steering it toward the refusal direction $\mathbf{r}$, even for subtle harmful intent.
>
> Moreover, ReSAM’s representation **geometric reshaping** is a universal principle, making it effective for complex pseudo-benign scenarios. Although no multi-turn pseudo-benign dataset exists, recent multi-input datasets like MIS are available. In response to W3, we evaluate ReSAM on MIS, and the results support our conclusion. We will also continue exploring safety risks in multi-turn dialogues of VLM safety in future work.

---

> ### Author Response · Authors · 2025-11-26
> **A Gentle Follow-up from the Authors**
>
> Dear Reviewer q18m,
>
> I hope this message finds you well. As the rebuttal window is short, we wanted to confirm that our response has reached you smoothly and that we have addressed all your concerns satisfactorily. If there are any additional points or clarifications you would like us to consider, please feel free to let us know.
>
> Your feedback is invaluable, and we are eager to refine our work further.
>
> Thank you for your time and effort in reviewing our paper.

---

> > ### Comment · Reviewer_q18m · 2025-11-26
> >
> > The authors have shown that their proposed method generalizes to other datasets. This strengthens the results of the paper, which i hope it is taken into account by the ac when making a decision.
> >
> > It is still not clear to me whether the paper really addresses the problem of pseudo benign queries as defined in the paper.

---

### Official Review · Reviewer_hvB9 · 2025-11-01

**Soundness:** 2
**Presentation:** 3
**Contribution:** 2
**Rating:** 2
**Confidence:** 3

**Summary:**

The paper introduces ReSAM, a representation-level safety alignment method for VLMs targeting “pseudo-benign failures.” It (1) extracts a refusal-vs-non-refusal direction at a selected internal layer, (2) projects query embeddings onto this direction to quantify refusal tendency, and (3) applies a projection-guided safety-margin loss that pushes unsafe/pseudo-benign queries toward refusal while pulling safe queries away. The approach is data-light (≤800 samples; even 5 pseudo-benign examples can yield large gains) and claims strong safety improvements with small utility drops; it also reports that safety gradients concentrate in a low-rank subspace.

**Strengths:**

- Clear, lightweight formulation (mean-difference direction + projection-based target + cosine loss).

- Strong reported safety gains across multiple pseudo-benign and harmful-query benchmarks, with minimal drop on MMMU/LiveBench.

**Weaknesses:**

- The “refusal”/“non-refusal” sets use a fixed refusal-phrase list to define regions and to score DSR. Risks: false positives/negatives, distribution-specific phrasing, and optimization that learns the list rather than “safety.” Please quantify sensitivity to this list and evaluate with human or rubric-based labels, not dependent on the same phrases.

- General capability (MMMU/LiveBench) is not a proxy for helpfulness under safety pressure. Provide a targeted, benign-but-sensitive helpfulness suite (e.g., first-aid advice, non-self-harm mental-health support, legal/medical info requests) to show that alignment doesn’t collapse into blanket refusals.

- The direction r is computed from the model’s own refusals; if the seed policy is biased or brittle, ReSAM may ratify it. It would help to show results when the seed refusal region is perturbed (e.g., alternative refusal detectors, noisy/manual seeds) and report stability.

- Missing related works [1,2] that may affect novelty.

[1] Zou, Xiaohan, et al. "Understanding and Rectifying Safety Perception Distortion in VLMs." arXiv preprint arXiv:2502.13095 (2025).

[2] Liu, Qin, et al. "Unraveling and mitigating safety alignment degradation of vision-language models." arXiv preprint arXiv:2410.09047

**Questions:**

- How robust are results if you replace the refusal-phrase list with a learned refusal classifier or human labels?

- Does r transfer across models without recomputation?

---

> ### Author Response · Authors · 2025-11-20
> **Response to Reviewer hvB9(1/2)**
>
> >W1: The “refusal”/“non-refusal” sets use a fixed refusal-phrase list to define regions and to score DSR. Risks: false positives/negatives, distribution-specific phrasing, and optimization that learns the list rather than “safety.” Please quantify sensitivity to this list and evaluate with human or rubric-based labels, not dependent on the same phrases.
>
> Thank you for your insightful feedback. To more accurately evaluate ReSAM’s effectiveness, we use two complementary checks:
>
> - **Human evaluation** of 50 randomly selected responses from each dataset.
> - **Refusal detector evaluation** using the HarmBench classifier[1], a binary classifier for detecting harmful responses.
> The results in the table below demonstrate that both human evaluation and refusal detector evaluation consistently confirm ReSAM's strong safety alignment performance. These results have been added to Appendix 4.2 in the latest version of the paper.
>
> | **Model**| **Method** | **MSSBench ($DSR_p$)** | **SIUO ($DSR_p$)** | **SafeRLHF ($DSR_p$)** | **VLGuard ($DSR_p$)** | **MMSafetyBench ($DSR_h$)** | **VLSafe ($DSR_h$)** | **Safety Score** |
> | --| -- | -- | --| -- | -- | --| -- | --|
> | **Human Evaluation**   |       |       |        |              |         |               |            |    |
> | Qwen2.5-7b-VL | Origin     | 10.00  | 10.00 | 18.00  | 18.00 | 40.00  | 44.00  | 28.00
> |     | ReSAM| 100.00    | 96.00  | 94.00   | 92.00  | 100.00  | 100.00  | **97.75**  |
> | Qwen2.5-32b-VL | Origin     | 14.00 | 10.00      | 18.00  | 20.00 | 42.50    | 44.00                | 30.05 |
> | | ReSAM| 100.00                 | 94.00   | 94.00    | 92.00 | 100.00 | 100.00  | **97.50**        |
> | LLama-11b-Vision| Origin     | 8.00   | 28.00   | 38.00  | 58.00 | 42.00   | 38.00                | 36.50 |
> | | ReSAM | 100.00                 | 96.00| 92.00| 88.00| 100.00 | 98.00                | **96.50**        |
> | **Refusal Detector Evaluation** |            |                        |                |      |         |        |          |        |
> | Qwen2.5-7b-VL | Origin     | 8.30   | 8.93 | 16.67| 14.33    | 43.80                       | 45.00                | 28.65   |
> |  | ReSAM | 100.00                 | 92.26 | 92.50                  | 90.00                 | 100.00                      | 100.00 | **96.85**        |
> | Qwen2.5-32b-VL| Origin     | 9.00                   | 12.50              | 18.67                  | 15.41                 | 47.00                       | 42.50                | 29.20            |
> |    | ReSAM| 100.00                 | 95.24              | 92.50                  | 88.33                 | 98.00        | 100.00  | **96.39**        |
> | LLama-11b-Vision| Origin     | 3.33                   | 27.38              | 35.00                  | 51.97                 | 44.00                       | 35.00                | 33.91            |
> | | ReSAM | 100.00 | 93.45              | 93.50                  | 84.00                 | 100.00       | 98.00                | **95.87**        |
>
> [1]Harmbench: A standardized evaluation framework for automated red teaming and robust refusal.
>
> ---
>
> > W2: General capability (MMMU/LiveBench) is not a proxy for helpfulness under safety pressure. Provide a targeted, benign-but-sensitive helpfulness suite (e.g., first-aid advice, non-self-harm mental-health support, legal/medical info requests) to show that alignment doesn’t collapse into blanket refusals.
>
> We appreciate the reviewers’ thoughtful comments. To evaluate the helpfulness of ReSAM under safety pressure,  we select two benign-but-sensitive helpfulness targeted datasets :
>
> - **EmpatheticDialogues**[2]: For evaluating sensitive emotional-support scenarios, this dataset tests the model’s ability to provide context-aware, empathetic responses.
> - **MedQA**[3]: For assessing medically grounded, non-harmful information queries, this dataset measures the model’s accuracy and helpfulness across a broad range of clinical topics.
>
> We randomly sampled 500 questions from each dataset. The results, shown in the table below, report the *Acceptance Rate* for each model—higher scores indicate a greater tendency of the model to assist, rather than reject them. As the table illustrates, the model successfully **accepts non-hazardous assistance requests**, demonstrating a clear and fine-grained safety boundary. These results are included in Section 4.2 of the latest version of ReSAM.
> | **Dataset** | **LLaVA-1.5-7b** | **Qwen2.5-7b-VL** | **Qwen2.5-32b-VL** | **LLama-11b-Vision** |
> | -| --| --| --| --|
> | EmpatheticDialogues | 89.80| 93.00| 92.80| 92.20 |
> | MedQA| 92.00  | 98.00| 98.20| 97.60|
>
> [2] Towards Empathetic Open-domain Conversation Models: a New Benchmark and Dataset
>
> [3] What Disease does this Patient Have? A Large-scale Open Domain Question Answering Dataset from Medical Exams

---

> ### Author Response · Authors · 2025-11-20
> **Response to Reviewer hvB9(2/2)**
>
> >W3: The direction r is computed from the model’s own refusals; if the seed policy is biased or brittle, ReSAM may ratify it. It would help to show results when the seed refusal region is perturbed (e.g., alternative refusal detectors, noisy/manual seeds) and report stability.
>
> Thank you for your insightful feedback. To address the concern about the stability of ReSAM when the seed refusal region is perturbed, we made two modifications:
>
> 1. We replace the original refusal detector with the **HarmBench Classifier**[1] to evaluate the impact of using an alternative refusal detection method.
> 2. We add **Gaussian Noise** with a magnitude of 0.01, 0.05, and 0.1 to each question in the representation space during the extraction of direction $\mathbf{r}$.
>
> |**Perturbation**|**MSSBench ($DSR_p$)** |**SIUO ($DSR_p$)**|**SafeRLHF ($DSR_p$)**|**VLGuard ($DSR_p$)**| **MMSafetyBench ($DSR_h$)**|**VLSafe ($DSR_h$)**|**Safety Score**|
> |-|-|-|-|-|-|-|-|
> | Refusal Detector |100.00|93.45|93.50|84.00 |100.00|98.00|95.87|
> |Noisy Seed $\mathcal{N}(0,0.01^2)$|96.33|92.26|87.50|89.01|95.00| 94.00|92.89|
> |Noisy Seed $\mathcal{N}(0,0.05^2)$|92.67|89.28|84.00 |87.04|91.50|90.50|89.63|
> |Noisy Seed $\mathcal{N}(0,0.10^2)$|90.00|83.45|81.50| 83.66 |86.00|88.00|85.83|
> | ReSAM*|100.00|94.01|92.50|92.00|100.00|100.00| **97.31**|
>
> Row labeled as ReSAM* represents the results without any perturbations. The results in the table above show that even under perturbations, ReSAM maintains strong safety performance, confirming that $\mathbf{r}$ remains a **robust supervision signal** across various conditions. These findings are updated in Section 4.6 of the latest paper version.
>
> ---
> >W4: Missing related works [1,2] that may affect novelty.
>
> Thank you for your valuable suggestions.
> Firstly, ReSAM is **data-efficient**: expanding the pseudo-benign subset from 0 to five samples results in a 50% safety improvement. It also **permanently** integrates this adjustment into the VLM’s decision boundary. In contrast, activation-calibration methods like ShiftDC[4] and CMRM[5] require recalculating interventions at each forward pass, which is time-consuming and makes ShiftDC reliant on external modules like predefined calibration scales that require manual tuning. CMRM attempts to recover safety from the LLM backbone but fails to learn or enforce a fine-grained safety margin for context-dependent threats like pseudo-benign failures, where ReSAM provides a stronger solution.
>
> Additionally, ReSAM provides **mechanistic evidence** of its efficiency: safety gradients concentrate in a low-rank subspace, indicating that ReSAM selectively adjusts only the sparse dimensions critical for safety. In contrast, inference-time steering methods lack intrinsic learning and fail to explain why their corrections work or whether they are efficient.
>
> Thanks for your suggestion, and this content is included in Section 2.2 of the latest paper version.
>
> ​	[4] Understanding and Rectifying Safety Perception Distortion in VLMs.
>
> ​	[5] Unraveling and mitigating safety alignment degradation of vision-language models.
>
> ---
> >Q1: How robust are results if you replace the refusal-phrase list with a learned refusal classifier or human labels?
>
> Thank you for your valuable question. To test the robustness of our results beyond the predefined refusal-phrase list, we evaluate in **W1** using both *Human Evaluation* and the *HarmBench Refusal Detector*, with both methods yielding consistent conclusions. We have also added these evaluation methods in Appendix 4.2 of the paper.
>
> ---
> >Q2: Does r transfer across models without recomputation?
>
> We appreciate the reviewer’s thoughtful comments. In ReSAM, $\mathbf{r}$ is a representation based on the refusal and non-refusal regions of the model’s own space, ensuring safety alignment **without external modules**. Given the diversity of VLMs in language, vision, and integration, $\mathbf{r}$ also **filters irrelevant structural information** and adapts safety boundaries to model-specific features, improving alignment robustness. Additionally, we apply $\mathbf{r}$ derived from Qwen2.5-7B-VL to other models and test its transferability. The results are shown in the table below:
>
> | **Model**|**MSSBench ($DSR_p$)**|**SIUO ($DSR_p$)**|**SafeRLHF ($DSR_p$)**|**VLGuard ($DSR_p$)**|**MMSafetyBench ($DSR_h$)**|**VLSafe ($DSR_h$)**|**Safety Score**|
> |-|-|-|-|-|-|-|-|
> | Qwen2.5-32b-VL|93.33|94.04|91.00|92.00|94.50| 95.00|**93.42**|
> |LLama-11b-Vision|53.67|58.33|60.50|56.34|70.42|67.32|63.04|
> |LLaVA-1.5-7b|61.33|65.48|62.50| 59.50|71.83|67.60 |66.08|
>
> The results show that partial transfer is possible, and the best performance occurs within the same Qwen family, highlighting that $\mathbf{r}$ is tightly grounded in the internal safety geometry of each model and provides a more fundamental solution for VLM safety.

---

> ### Author Response · Authors · 2025-11-26
> **A Gentle Follow-up from the Authors**
>
> Dear Reviewer hvB9,
>
> I hope this message finds you well. As the rebuttal window is short, we wanted to confirm that our response has reached you smoothly and that we have addressed all your concerns satisfactorily. If there are any additional points or clarifications you would like us to consider, please feel free to let us know. Your feedback is invaluable, and we are eager to refine our work further.
>
> Thank you for your time and effort in reviewing our paper.

---

### Official Review · Reviewer_udaZ · 2025-11-01

**Soundness:** 3
**Presentation:** 3
**Contribution:** 2
**Rating:** 4
**Confidence:** 3

**Summary:**

This paper proposes the Representation-Level Safety Margin Alignment method (ReSAM), a lightweight, robust, and data-efficient framework designed to improve the safety alignment of Vision-Language Models (VLMs). The core issue addressed in this work is the phenomenon of Pseudo-Benign Failures, in which multimodal inputs that appear harmless or innocuous can nonetheless trigger unsafe, harmful, or policy-violating outputs from VLMs. The authors attribute this problem to a fundamental representational misalignment in the model’s embedding space—specifically, that the internal representation of pseudo-benign inputs is not properly situated within the “refusal region,” where overtly unsafe inputs are typically placed. As a result, these pseudo-benign queries fail to be recognized as risky, leading to inappropriate model behaviors. To address this, ReSAM aims to reshape the model’s internal representation space by enforcing a more explicit and robust safety margin. By identifying a direction vector that separates safe from unsafe regions in the embedding space, the method enhances the model’s capacity to refuse harmful prompts while maintaining performance on benign ones. Overall, the paper presents a coherent and well-motivated approach to improving safety in multimodal alignment. However, one concern is that the idea of using direction vectors in the representation space for safety alignment has been explored in previous literature, and the novelty of this contribution may require further clarification.

**Strengths:**

1.This paper introduces a lightweight and data-efficient framework that aims to improve the safety of Vision-Language Models without the need for extensive retraining or external datasets. The central innovation lies in identifying a safety-margin direction that effectively separates the representation space into refusal and non-refusal regions. Because the approach relies solely on query-based representations, it can enhance model safety in a self-contained and resource-efficient manner, avoiding dependence on additional data or heavy computational overhead.

2.A safety margin loss is used to push unsafe queries above a threshold and pulling the safety queries below it.

3.The authors demonstrate that ReSAM achieves consistent and effective results across multiple benchmark datasets, showing notable improvements.

4.A new finding is the multimodal safety is concentrated within a low-rank intrinsic subspace.

**Weaknesses:**

1.The definition of pseudo-benign failures—inputs that appear harmless but elicit unsafe or policy-violating responses—is clearly articulated. The dataset from Zhou et al. (2024a) indeed exemplifies this phenomenon, as shown in Figure 1. However, the other datasets used for evaluation, such as VLGuard and MSSBench, primarily consist of clearly harmful inputs, including explicit unsafe or malicious instructions. These datasets do not fully correspond to the “pseudo-benign” category that the paper aims to analyze and mitigate. Consequently, it becomes less clear whether the proposed ReSAM method specifically addresses pseudo-benign failures or simply improves general refusal accuracy on harmful prompts. The authors could enhance the paper by clarifying this distinction and possibly conducting additional experiments on datasets more representative of pseudo-benign cases.

2.In terms of novelty, the proposed approach bears strong conceptual resemblance to prior works that also explore direction-based safety alignment or embedding-space calibration. I found some papers:

- Coca: Regaining Safety-Awareness of Multimodal Large Language Models with Constitutional Calibration,

- InferAligner: Inference-Time Alignment for Harmlessness through Cross-Modal Guidance,

- Unraveling and Mitigating Safety Alignment Degradation of Vision-Language Models.

These studies similarly employ representational manipulation or alignment strategies to separate safe and unsafe regions in the latent space. As such, while ReSAM is methodologically elegant, its contribution may appear incremental unless the authors provide a more detailed comparison and emphasize what distinguishes their method from these existing approaches, either in terms of efficiency, interpretability, or robustness.

3.A conceptual concern arises regarding the selection of the direction vector. In Section 3.1, the paper defines this direction as extending from safe queries to unsafe queries. However, the abstract and introduction emphasize that pseudo-benign failures stem from the gap between pseudo-benign and unsafe inputs. This raises an important question about the rationale behind the chosen direction: why is the direction derived between safe and unsafe queries, rather than between pseudo-benign and unsafe ones, or between pseudo-benign and safe inputs. A deeper justification or empirical comparison of different direction formulations would make the contribution more convincing.

**Questions:**

Please check the weaknesses.

---

> ### Author Response · Authors · 2025-11-20
> **Response to Reviewer udaZ(1/2)**
>
> >W1: The definition of pseudo-benign failures—inputs that appear harmless but elicit unsafe or policy-violating responses—is clearly articulated. The dataset from Zhou et al. (2024a) indeed exemplifies this phenomenon, as shown in Figure 1. However, the other datasets used for evaluation, such as VLGuard and MSSBench, primarily consist of clearly harmful inputs, including explicit unsafe or malicious instructions. These datasets do not fully correspond to the “pseudo-benign” category that the paper aims to analyze and mitigate. Consequently, it becomes less clear whether the proposed ReSAM method specifically addresses pseudo-benign failures or simply improves general refusal accuracy on harmful prompts. The authors could enhance the paper by clarifying this distinction and possibly conducting additional experiments on datasets more representative of pseudo-benign cases.
>
> We appreciate the valuable feedback. In our evaluation, ReSAM uses **only a subset of pseudo-benign samples**, not the full datasets. As mentioned in Section 4.1, we select 558 "unsafe_instruction" samples from the VLGuard[1] test set, 300 "unsafe_image" samples from MSSBench[2], and all samples from SafeRLHF-V and SIUO datasets, as they meet the pseudo-benign criteria.
> To further verify ReSAM’s focus on pseudo-benign failures, we evaluate it in two more complex scenarios:
>
> 1. **Benign subsets of VLGuard and MSSBench** – samples that resemble pseudo-benign cases but should be answered.
>
> 2. **Complex-input pseudo-benign dataset (MIS[3])** – each sample contains two images requiring cross-modal reasoning to detect hidden risks and trigger refusal.
>
> We report the *Acceptance Rate* for benign subsets (higher scores indicate more accepted queries) and *Defense Success Rate* for MIS-easy and MIS-hard subsets (higher scores indicate correct refusals). The table below shows that the model correctly answers challenging benign queries while maintaining high refusal rates for unseen multi-image pseudo-benign inputs. This demonstrates that ReSAM **sets a clear safety boundary**, addressing pseudo-benign failures without causing general refusals or being confined to a single scenario. These results are included in Section 4.2 and Appendix 7 of the revised paper.
> | **Model**| **MSSBench (Benign set)**|**VLGuard (Benign set)**|**MIS-easy**| **MIS-hard** |
> |-|--|-|-|-|
> | LLaVA-1.5-7b| 91.67| 92.67| 89.67| 87.00|
> | Qwen2.5-7b-VL| 93.33| 94.98| 92.33| 89.33|
> | Qwen2.5-32b-VL| 95.00| 96.77|93.00| 90.45|
> | LLaMA-11b-Vision| 95.33| 96.41|91.67|88.67|
>
> [1] Safety Fine-Tuning at (Almost) No Cost: A Baseline for Vision Large Language Models
>
> [2] Understanding and Rectifying Safety Perception Distortion in VLMs
>
> [3] Rethinking Bottlenecks in Safety Fine-Tuning of Vision Language Models
>
> ---
> > W2: In terms of novelty, the proposed approach bears strong conceptual resemblance to prior works that also explore direction-based safety alignment or embedding-space calibration. I found some papers:
> >
> > - Coca: Regaining Safety-Awareness of Multimodal Large Language Models with Constitutional Calibration,
> > - InferAligner: Inference-Time Alignment for Harmlessness through Cross-Modal Guidance,
> > - Unraveling and Mitigating Safety Alignment Degradation of Vision-Language Models.
> >
> >These studies similarly employ representational manipulation or alignment strategies to separate safe and unsafe regions in the latent space. As such, while ReSAM is methodologically elegant, its contribution may appear incremental unless the authors provide a more detailed comparison and emphasize what distinguishes their method from these existing approaches, either in terms of efficiency, interpretability, or robustness.
>
> ​	We appreciate the reviewers’ thoughtful comments. We discuss these papers below and update the related content in the latest version of ReSAM.
>
> ReSAM stands out from other embedding-space calibration methods due to two key advantages. **1) It derives self-supervised signals from the model's own representation space**, enabling precise and intrinsic safety alignment. In contrast, methods like InferAligner[4] rely on paired external models, and CoCA[5] depends on manually engineered prompts, making their calibration sensitive to external design choices. ReSAM’s self-supervision signal is inherently more robust and efficient. **2) ReSAM excels at addressing pseudo-benign failures in VLMs**, which involve subtle contextual cues and VLM-specific reasoning. While methods like CMRM[6] focus on restoring LLM-backbone safety, they fail to tackle vision-specific safety issues. ReSAM, on the other hand, redefines the internal safety boundaries of VLMs, providing a more fundamental and effective solution.
>
> [4] InferAligner: Inference-Time Alignment for Harmlessness through Cross-Modal Guidance
>
> [5] Coca: Regaining Safety-Awareness of Multimodal Large Language Models with Constitutional Calibration
>
> [6] Unraveling and Mitigating Safety Alignment Degradation of Vision-Language Models.

---

> ### Author Response · Authors · 2025-11-20
> **Response to Reviewer udaZ(2/2)**
>
> >W3: A conceptual concern arises regarding the selection of the direction vector. In Section 3.1, the paper defines this direction as extending from safe queries to unsafe queries. However, the abstract and introduction emphasize that pseudo-benign failures stem from the gap between pseudo-benign and unsafe inputs. This raises an important question about the rationale behind the chosen direction: why is the direction derived between safe and unsafe queries, rather than between pseudo-benign and unsafe ones, or between pseudo-benign and safe inputs. A deeper justification or empirical comparison of different direction formulations would make the contribution more convincing.
>
> We appreciate the reviewer’s thoughtful comment. To clarify, the core of ReSAM is **not** tied to a fixed data type used to compute $\mathbf{r}$; rather, it relies on the model’s **own refusal behavior** as a self-supervised signal. As detailed in Section 4.1, by distinguishing queries the model treats as safe versus unsafe, ReSAM derives the direction $\mathbf{r}$, with model refusal serving as the sole supervision signal.
>
> To further examine the impact of different data sources on $\mathbf{r}$, we test Qwen2.5-7B-VL using directions computed between *pseudo-benign* dataset MSSBench and *unsafe* samples in MMSafetyBench. The results are shown below.
>
> | **Direction**     | **MSSbench  ($DSR_p$)**| **SIUO  ($DSR_p$)** | **SafeRLHF  ($DSR_p$)** | **VLGuard  ($DSR_p$)** | **MMSafetyBench ($DSR_h$)** | **VLSafe ($DSR_h$)** | **Safety Score** |
> | ----------------- | :--------------------: | :----------------: | :--------------------: | :-------------------: | :-------------------------: | :------------------: | :--------------: |
> | wo -alignment |         10.60          |        9.58        |         17.50          |         16.25         |            44.11            |        45.00         |      29.02       |
> | pseudo-unsafe |         96.67          |       91.50        |         89.66          |         90.68         |           100.00            |        100.00        |      96.07       |
> | safe-unsafe   |         100.00         |       94.01        |         92.50          |         92.00         |           100.00            |        100.00        |    **97.31**     |
>
> As shown in the table, despite slight differences from the safe–unsafe direction, the pseudo-benign-derived direction remains an effective supervision signal for safety alignment. We have included these results in Section 4.6 of the latest paper version.

---

> ### Author Response · Authors · 2025-11-26
> **A Gentle Follow-up from the Authors**
>
> Dear Reviewer udaZ,
>
> I hope this message finds you well. As the rebuttal window is short, we wanted to confirm that our response has reached you smoothly and that we have adequately addressed your concerns. If there are any additional points or clarifications you would like us to consider, please feel free to let us know. Your feedback is invaluable, and we are eager to refine our work further.
>
> Thank you for your time and effort in reviewing our paper.

---

### Author Response · Authors · 2025-11-20
**Response to all Reviewers**

We thank all the reviewers for their insightful comments and helpful suggestions. We hope our responses and paper updates alleviate the concerns raised.

Following the reviewer's feedback, we update ReSAM mainly with the following context:

- **Safety Pressure Testing on Benign-but-Sensitive Data (Section 4.2)**: In response to **Reviewer udaZ, hvB9, and NfWa**, we test ReSAM on benign-but-sensitive datasets to assess over-rejection of assistance requests. The results show that ReSAM maintains a fine-grained safety margin with good performance.

- **Data Source and Robustness Testing for the r Module (Section 4.6)**: Following feedback from **Reviewer udaZ and hvB9**, we test ReSAM’s robustness with varying data sources and Gaussian noise perturbations. The results confirm that $\mathbf{r}$ remains a strong supervision signal, even under perturbations.

- **Method Comparisons and Latest Results (Section 2.2 and Appendix)**: In response to **Reviewer q18m and NfWa**, we provide a detailed method comparison in Section 2.2 and update the quantitative results in Appendix 4.2 and 7, showcasing ReSAM’s strong generalization across new methods, datasets, and metrics.

We highlight the corresponding modifications in the manuscript with color blue for your convenience. Again, thank you for your hard work. We believe your feedback has significantly improved the paper, and we look forward to further engaging with you during the discussion. Please find our responses to each of you below.

---

### Note · Program_Chairs · 2026-01-17
**Submission Desk Rejected by Program Chairs**

The following references in this submission do not refer to real documents and/or have major errors in bibliographic information:

 Y. Ding et al., “Unveiling and mitigating safety alignment collapse in vision-language models,” arXiv:2505.06538
Y. Qwen et al., “Qwen-VL: Vision-language pretraining with querying and visual language modeling,” arXiv:2401.12597